# META-LEARNING WITH INDIVIDUALIZED FEATURE SPACE FOR FEW-SHOT CLASSIFICATION

## ABSTRACT

Meta-learning provides a promising learning framework to address few-shot classification tasks. In existing meta-learning methods, the meta-learner is designed to learn about model optimization, parameter initialization, or similarity metric. Differently, in this paper, we propose to learn how to create an individualized feature embedding specific to a given query image for better classifying, i.e., given a query image, a specific feature embedding tailored for its characteristics is created accordingly, leading to an individualized feature space in which the query image can be more accurately classified. Specifically, we introduce a kernel generator as meta-learner to learn to construct feature embedding for query images. The kernel generator acquires meta-knowledge of generating adequate convolutional kernels for different query images during training, which can generalize to unseen categories without fine-tuning. In two standard few-shot classification data sets, i.e. Omniglot, and *mini*ImageNet, our method shows highly competitive performance.

## 1 INTRODUCTION

A well-performing model for image classification is often equipped with a discriminative feature space and a powerful classifier, which heavily rely on training with large quantities of labeled data, especially for those high-capacity deep models. The scalability of most deep models to new tasks with few samples is severely limited because of the dependence on big data. In contrast, humans excel at recognizing objects and can rapidly learn to recognize a new class even with a single example by utilizing prior knowledge Lake et al. (2011). The significant gap between human and machine learning motivates the development of one/few-shot learning that aims at achieving better generalization in the tasks with scarce labeled data.

The task of few-shot classification is to recognize previously unseen classes with very few labeled examples. A straightforward approach is to fine-tune a pre-trained model on the new classes, as is usually done in transfer learning Finn et al. (2017). But it becomes awkward in few-shot classification, as it would cause severe over fitting when only a few training examples are available. Data augmentation and regularization techniques can alleviate the over fitting problem Sung et al. (2018), but they are still far from obtaining a satisfactory model in few-shot scenarios. One promising approach to address few-shot classification is meta-learning Thrun (1998); Hochreiter et al. (2001), in which transferable knowledge is learned on a set of tasks. The idea is that by learning on task $\tau$ following a distribution $p(\tau)$, a meta-learner may figure out an effective learning strategy tailored for the task distribution $p(\tau)$. Therefore, a meta-learner is able to quickly adapt to new tasks following the same distribution $p(\tau)$ with only a few examples per class.

Meta-learner plays a crucial role in meta-learning. Generally, a meta-learner is a model designed to acquire learning algorithms (meta-knowledge), which are used to further teach a learner so that the learner can quickly adapt to new tasks. In existing meta-learning methods, the meta-learner is designed as an optimizer Andrychowicz et al. (2016); Ravi & Larochelle (2017); Li & Malik (2017), an initializer Finn et al. (2017); Li et al. (2017); Nichol et al. (2018), or as a kind of distance metric Vinyals et al. (2016); Koch et al. (2015); Snell et al. (2017); Sung et al. (2018). A meta-learner for optimization usually learns to predict gradients of parameters or learns SGD-like parameter updating policy to make learners converge quickly. A meta-learner that learns to initialize model parameters aims to initialize a learner so that it can adapt to new tasks with a small amount of data in a few steps. In those metric-learning based methods, the meta-learner is often designed as a kind

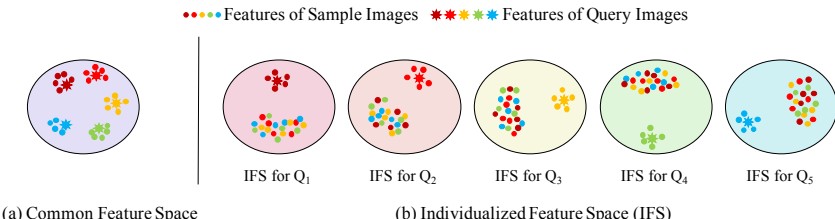

(a) Common Feature Space     (b) Individualized Feature Space (IFS)

**Figure 1.** Comparison of common feature space and individualized feature space. (a) The common feature space. Feature distribution has to be discriminative enough so that all classes can be distinguished from each other. (b) The individualized feature space. There is an individualized feature space tailored for each query image (i.e. the image to be classified), where images are classified into two categories according to whether they belong to the same class as the query image.

of non-parametric distance metric or deep distance metric implemented with a deep model, which are expected to generalize to unseen classes after being trained on the training set.

Different from existing works, in this paper, we propose to learn how to create an individualized feature embedding specific to a given query image for more accurately classifying during meta-learning. To classify a query image, we need to compare it with labeled images of all classes. In general, as different query images own different characteristics because of inter-class variations or intra-class variations (e.g. category, pose, illumination, and some other factors), one tends to focus on different aspects of those labeled images when comparing them to different query images. Thus, the characteristics of query images influence the perception of labeled images. Inspired from this, our meta-learner is designed to learn how to create feature embedding tailored for characteristics of different query images, so that the query image can be more accurately classified.

The construction of individualized feature spaces naturally leads to formulation of binary classification tasks. Most existing methods map all images into a common feature space. In such situation, to distinguish different classes from each other, images are expected to form multiple clusters according to their class labels, leading to a multi-class classification task. By contrast, when an individualized feature space is constructed specifically for a given query image in our method, one only needs to distinguish the class which the query image belongs to from the rest of the classes, without concerning whether the rest of the classes can be distinguished from each other. Therefore, the goal of our meta-learner is to learn how to create individualized feature embedding tailored for the query image that will minimize (or maximize) the distances between the query image and those belonging to the same class (or different classes). Figure 1 illustrates the differences between the multi-class classification task in a common feature space and the binary classification tasks in our individualized feature spaces.

Specifically, the meta-learner in our proposed method is designed as a kernel generator to construct the feature embedding for a specific query image. It generates distinct sets of convolutional kernels as the characteristics of query images vary. By convolving features of images with those generated kernels, they are mapped into the individualized feature space constructed with the corresponding query image. During meta-training, the learning of the kernel generator is at higher level than that of ordinary network modules. It acquires meta-knowledge of generating adequate convolutional kernels for different query images, which can generalize to new classes without fine-tuning.

Overall, our contributions lie in three folds. **First**, we propose to learn how to create an individualized feature embedding tailored for a given query image. By considering the characteristics of the query image in feature embedding, an individualized feature space is constructed to accurately classify the query image. **Second**, we offer an effective implementation of the meta-learner as a kernel generator, which learns to generate convolutional kernels based on the given query image. **Third**, on two standard few-shot classification data sets, including Omniglot Lake et al. (2011), and *mini*ImageNet Vinyals et al. (2016), our method achieves competitive results.

## 2 RELATED WORK

**Meta learning** Meta-learning approaches Thrun (1998); Schmidhuber (1987); Naik & Mammone (1992) involve training a meta-learner to acquire meta-knowledge, which generally can be trans-

ferred into new tasks with scarce data and within a few iterations Andrychowicz et al. (2016); Schmidhuber et al. (1997).

Some works design the meta-learner as an optimizer that learns to update model parameters Bengio et al. (1992); Schmidhuber (1992). Similar method has been applied to deep networks to compose gradients of parameters for quick convergence of training Hochreiter et al. (2001); Andrychowicz et al. (2016); Li & Malik (2017). Recently, Ravi & Larochelle (2017) proposes a meta-learner based on LSTM Hochreiter & Schmidhuber (1997) to learn a SGD-like parameter updating policy which is used to train another learner in the few-shot regime. It also learns a general initialization of the learner network to accelerate its convergence.

Meta-learner learning how to initialize a learner is more general and generic in meta-learning. Finn et al. (2017) proposes an algorithm, which is referred to as Model-Agnostic Meta-Learning (MAML), learning how to initialize a learner so that the learner can adapt to new task with a small amount of training data in just a few steps. MAML is compatible with any model trained with gradient descent. Li et al. (2017) improves MAML with not only learning the learner initialization, but also the learner update direction and learning rate. Nichol et al. (2018) analyzes the first-order MAML, and points out it is simpler to implement than was widely recognized prior, forming a new algorithm, called Reptile.

Metric-learning methods design the meta-learner as a kind of distance metric. Generally, those approaches attempt to learn a common feature space where categories can distinguish from each other based on the defined or learned distance metric, forming a multi-class classification task. Our work is related to metric learning. The difference is that our meta-learner goals for learning how to create an individualized feature space tailored for a given query image, where we only need to distinguish class the query image belongs to from all other classes, forming a binary classification task. More clear comparison can be seen in Figure 1. Koch et al. (2015) formulates the one-shot classification task as the matching problem and train Siamese neural networks to calculate the similarity between images in the support set and a query image. Vinyals et al. (2016) proposes Matching Networks where a fully differentiable neural attention mechanism is applied into nearest neighbor classifier to classify a query example with the support set. Prototypical Networks Snell et al. (2017) learn the prototype feature vector for each class as the average of all feature vectors extracted by images of the class in sample set, with which the distance between a query image and a class can be computed. Instead of defining the non-parametric distance metric, Relation Network Sung et al. (2018) learns to learn a deep distance metric by a sub-network.

Some other meta-learning approaches involve training a generic neural architecture. Santoro et al. (2016) trains a memory-augmented LSTM for few-shot learning, where the learner is trained to adapt to new tasks as the LSTM rolls out. Mishra et al. (2017) proposes a temporal convolutional network that outputs the prediction for the test example based on the previous labeled examples it has seen given as input a sequence of example-label pairs followed by an unlabeled example.

**Parameter Prediction** Our work is also related to parameter prediction. To accelerate the convergence rate of model, Ha et al. (2017) proposes to generate the weights of main network via a HyperNetwork with fewer learnable parameters, which can be viewed as a relaxed form of weight-sharing across layers. The difference between HyperNetwork and our work is that, for a CNN, parameters of the main network are fixed after training in HyperNetwork, while kernels are generated dynamically conditioned on input query images even when testing in our method. De Brabandere et al. (2016) and Klein et al. (2015) apply input conditioned kernel to image prediction task, and the dynamically-generated filters are mainly used to predict the movement of pixels between frames. Han et al. (2018) proposes the contrastive convolutional kernels that is created based on the input face pair to focus on the different features between the input face pair for better certifying whether they belong to the same identity. Differently, our dynamic generated kernels corresponding to a given query image are used to build a individualized feature space, where the query image can be more accurately classified. Qiao et al. (2018) and Gidaris & Komodakis (2018) propose to learn how to generate the classification weight vectors given the feature presentations of a few images of a specific class to address the few-shot classification task. Different from them, our meta-learner learns how to map examples into the individualized feature space corresponding to a given query image. Our work is related to Bertinetto et al. (2016) the most. Bertinetto et al. (2016) proposes a second network, called a *Learnet*, to predict the parameters of a pupil network from a exemplar. The

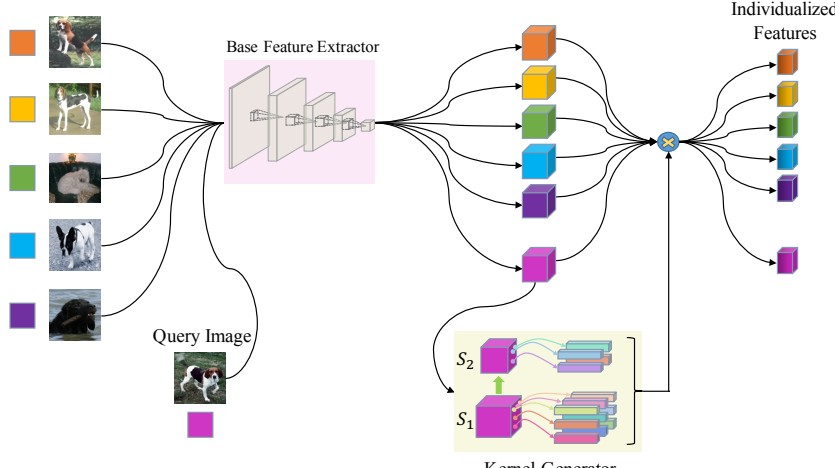

**Figure 2.** Network architecture for a 5-way 1-shot classification task.

novelty of our work is still evident relative to *Learnet*. First, the mechanisms of generating kernels are different. *Learnet* predicts kernels by directly reshaping the output of the last layer of *Learnet* into the target shape, possibly conflicting with the receptive field. Differently, to preserve the spatial information, our kernel generator creates distinct kernels focusing on different components of the input image. Second, the structure design of the backbone of the two methods is different. *Learnet* aims to dynamically generate the whole network, i.e., the backbone of each input is different from each other, which is unnecessary and makes it hard to optimize. In contrast, as we believe that the low-level features of different inputs are generic, the backbone in our method is shared by all inputs and only the high-level features are dynamic.

## 3 INDIVIDUALIZED FEATURE SPACE

### 3.1 TASK DEFINITION

Generally, there are three sets of examples in a few-shot classification task: a training set, a support set, and a testing set. The training set and the support set have disjoint label spaces with each other, while the testing set shares the same label space with the support set. The few-shot classification task is to classify examples in the testing set with a small support set, i.e. there are only one or a few examples for each class. A $C$-way $K$-shot classification task means there are $C$ classes with $K$ labeled examples for each class in the support set.

In this work, we follow the episode-based training strategy, and perform one-shot ($K$=1) and five-shot ($K$=5) classification in our experiments. The episode-based training scheme is an effective training scheme in meta-learning, which is widely used in Snell et al. (2017); Finn et al. (2017); Li et al. (2017); Sung et al. (2018); Mishra et al. (2017). It simulates the support/test set split that is used at testing phase to design the episode as a sample/query set split. The sample set consists of $C$ classes and $K$ labeled samples per class, which are randomly selected from the training set, and the query set consists of a fraction of the remainder of samples of the $C$ classes.

### 3.2 FRAMEWORK

Suppose there is a sample set $\{(x_{ij}, y_{ij})\}_{i \in \{1,...,C\}, j \in \{1,...,K\}}$ and a query image $q$ with label $l$, where $x_{ij}$ means the $j$-th image of the $i$-th class in the sample set, and $y_{ij}$ is the label of image $x_{ij}$. As is shown in Figure 2, the framework of our method mainly consists of two modules: a base feature extractor and a kernel generator. The base feature extractor $c$, which is shared across all images, consisting of several cascaded convolutional layers, extracts shared base features for all input images.

Taking $x$ as one image in the sample set, as well as $F_x$ and $F_q$ as feature maps of image $x$ and $q$ respectively, $c$ can be formulated as:

$$F_x = c(x), F_q = c(q) \in \mathbb{R}^{h_F \times w_F \times c_F} \tag{1}$$

where $h_F$, $w_F$ and $c_F$ are the height, width, and number of channels of output feature maps respectively. At this step, images are mapped from the raw image space to a common feature space.

To better distinguish the query image from other objects, the individualized feature space is created based on the base feature space, where only characteristics that can best distinguish the query image from other objects are focused. Specifically, a kernel generator $G$ is introduced to generate individualized kernels for different query images. The kernel generator $G$ takes the feature maps $F_q$ as input, and outputs a set of individualized kernels $K_q$ for the query image $q$, which can be generally formulated as follow:

$$K_q = G(F_q) \tag{2}$$

The set of individualized kernels $K_q$ of image $q$ can be used to create a query specific feature embedding: $f^q : S^{Com} \longrightarrow S_q^{Ind}$, where the $S^{Com}$ and $S_q^{Ind}$ means common feature space corresponding to the base feature extractor and individualized feature space of $q$ respectively. $f^q$ is constructed as the form of convolution with $K_q$ as convolutional kernels, that is:

$$f^q(F_x) = K^q * F_x \triangleq F_x^q, f^q(F_q) = K^q * F_q \triangleq F_q^q \tag{3}$$

here, * means the conventional convolution. Thus, $f^q$ is a dynamic feature embedding, and varied with the query image.

An individualized feature space of image $q$ is the range of function $f^q$. In the individualized feature space $S_q^{Ind}$, the distance between images in the sample set and image $q$ can be defined as any distance metric, such as cosine distance, Manhattan distance, and Euclidean distance. Based on the distance metric we used, images in the sample set are expected to gather into two clusters: images owning the same class label as $q$, and images with different labels from $q$.

## 3.3 KERNEL GENERATOR

Convolutional kernels generated via the kernel generator $G$ should be able to obtain adequate information of the given query image to create expressive individualized feature space. Inspired by Han et al. (2018), our kernel generator is designed as a hierarchical structure to obtain kernels with multiple scales as feature maps in different layers usually own different receptive field. Specifically, there are $T$ layers in kernel generator network, one sub-generator for each layer, forming $T$ sub-generators in total:

$$G = \{g_1, g_2, \cdots, g_T\}. \tag{4}$$

Layer $i(1 \leq i \leq T)$ starts from feature maps $S_i^q$, which are usually obtained by operating the convolution on the feature maps $S_q^{i-1}$ with $S_q^0 = F_q$.

On each layer, a sub-generator $g_i$ is constructed to generate a group of kernels in the same scale as below:

$$K_q^i = \{k_q^{i1}, k_q^{i2}, ..., k_q^{iN_i}\}, \tag{5}$$

where $N_i$ is the number of kernels generated from $g_i$. Each kernel $k_q^{ij}$ is expected to portray the characteristics of a local component of image $q$, achieved by cropping a local patch:

$$K_q^{ij} = g_i(P_q^{ij}), P_q^{ij} = R(F_q^i, c_{ij}, h, w), \tag{6}$$

where $R$ denotes the image crop operation and $R(F_q^i, c_{ij}, h, w)$ means cropping $F_q^i$ with the center at $c_{ij}$, height of $h$, and width of $w$. In our experiments, $g_i$ consists only one fully connected layer.

The kernels from one sub-generator share similar receptive field but focus on different components. Kernels from different sub-generators have different receptive fields paying attention to characteristics in different scales. Altogether, a set of individualized kernels can be obtained as the union of kernels from all the sub-generators as below:

$$K_q = \{k_q^{11}, ..., k_q^{1N_1}, ..., k_q^{ij}, ..., k_q^{T1}, ..., k_q^{TN_T}\}. \tag{7}$$

The learning of kernel generator is higher level than vanilla kernel, acquiring meta-knowledge of generating adequate convolutional kernels for different query images during training, which can generalize to unseen categories without fine-tuning.

## 3.4 Loss Function

Given a query image $q$ with label $l$ in a query set of an episode, we hope features of images in the sample set to be close to or far from the features of $q$ in the individualized feature space according to whether they have the same label with $q$. The possibility $p_l^q$ that label $l$ is assigned to image $q$ can be calculated by a softmax function based on the defined distance metric. Following the cross entropy loss, the loss function is formulated as follow:

$$L_1 = -\frac{1}{N} \sum_{(q,l)} log(p_l^q) \tag{8}$$

where $N$ is the number of query images. Minimizing the above loss function can guarantee that the true label $l$ be assigned to the query images $q$. Moreover, individualized kernels are expected to capture the intrinsic characteristics of a object, that is, individualized kernels of different images with the same object should be the same even with different poses, illuminations, forming another cross entropy loss:

$$L_2 = -\frac{1}{N} \sum_{(q,l)} e_l log(H(K_q)) \tag{9}$$

where $e_l$ is a one-hot vector with 1 in the $l-$th position, and $H(K_q)$ aims to regress the possibility distribution of kernels $K_q$ to a one-hot code for classification. The network can be trained in an end-to-end manner by jointly optimizing $L_1$ and $L_2$ with the gradient decent based optimization algorithm.

## 4 Experiments

### 4.1 Settings

We implement our method with PyTorch. In all experiments, the parameters of our models are randomly initialized, and Adam Kinga & Adam (2015) is used for optimization. Models are iterated 100k with learning rate 0.0005 during training. The batch size is set as 8. As for the kernel generator, it consists of two sub-generators. The first sub-generator takes $6 \times 6$ feature maps as input, generating 36 kernels, and the second one takes $4 \times 4$ feature maps as input, generating 16 kernels. For a $K$-shot task, we average the features of $K$ images with the same label in the individualized feature space tailored for a given query image, and then calculate the distance between the query image and the mean features as is done in Snell et al. (2017).

As most few-shot learning models utilize four convolutional blocks for feature embedding module Vinyals et al. (2016); Snell et al. (2017); Finn et al. (2017); Li et al. (2017); Sung et al. (2018), the base feature extractor in our model is also designed as a convolution architecture with 4 modules, where each module consists of a $3 \times 3$ convolution layer with 64 filters, followed by batch normalization Ioffe & Szegedy (2015), a ReLU nonlinearity, and $3 \times 3$ max-pooling. The stride for all convolution layers is 1. On Omniglot, step size in the first two max-pooling layers is 2, and in the last two max-pooling layers is 1 to make sure the network outputs bigger feature map, while on *mini*ImageNet, all step size of max-pooling layers is 2 due to the increased image size. We refer the network described above to as $C_4$. Besides, as Mishra et al. (2017) uses the small version of the ResNet He et al. (2016) to improve the accuracy, we also use a similar ResNet structure but with less filters on *mini*ImageNet, which is referred to as ResNet.

For all experiments, on a $C$-way $K$-shot experiment, the episode is formed with $C$ classes, and each class includes $K$ sample images, and 6 and 15 query images for training and testing respectively. Following Vinyals et al. (2016); Snell et al. (2017); Finn et al. (2017); Sung et al. (2018), on Omniglot, we compute few-shot classification accuracies by averaging over 1000 randomly generated episodes from the testing set, while on *mini*ImageNet and *tiered*ImageNet, results are computed by averaging over 600 randomly generated episodes from the testing set.

**Table 1.** Few-shot classification accuracies on Omniglot. '-': not reported.

| Model | Fine Tune | 5-way accuracy(%) | | 20-way accuracy(%) | |
|---|---|---|---|---|---|
| | | 1-shot | 5-shot | 1-shot | 5-shot |
| MANN Santoro et al. (2016) | N | 82.8 | 94.9 | - | - |
| Convolutional Siamese Nets Koch et al. (2015) | N | 96.7 | 98.4 | 88.0 | 96.5 |
| Convolutional Siamese Nets Koch et al. (2015) | Y | 97.3 | 98.4 | 88.1 | 97.0 |
| Matching Nets Vinyals et al. (2016) | N | 98.1 | 98.9 | 93.8 | 98.5 |
| Matching Nets Vinyals et al. (2016) | Y | 97.9 | 98.7 | 93.5 | 98.7 |
| Siamese Nets with Memory Hertwig et al. (2004) | N | 98.4 | 99.6 | 95.0 | 98.6 |
| Neural Statistician Edwards & Storkey (2017) | N | 98.1 | 99.5 | 93.2 | 98.1 |
| Meta Nets Munkhdalai & Yu (2017) | N | 99.0 | - | 97.0 | - |
| Prototypical Nets Snell et al. (2017) | N | 98.8 | 99.7 | 96.0 | 98.9 |
| MAML Finn et al. (2017) | Y | 98.7 | **99.9** | 95.8 | 98.9 |
| Meta-SGD Li et al. (2017) | Y | **99.53** | **99.93** | 95.93 | 98.97 |
| Relation Net Sung et al. (2018) | N | **99.6** | **99.8** | 97.6 | 99.1 |
| SNAIL Mishra et al. (2017) | Y | 99.07 | 99.78 | 97.64 | **99.36** |
| ours | N | **99.49** | **99.83** | **98.03** | **99.22** |

## 4.2 OMNIGLOT

Omniglot Lake et al. (2011) consists of 1623 characters collected from 50 different alphabets. Each character contains 20 samples drawn by different people. Following Santoro et al. (2016); Finn et al. (2017); Snell et al. (2017), we resize all images into $28 \times 28$, and augment new classes through randomly rotating images by 90 degrees multiples times. We randomly choose 1200 classes for meta-training and the remaining 423 classes for meta-testing, both with rotation augmentation.

Results on Omniglot is shown in Table 1. As can be seen, accuracies of most methods approach $100\%$, which means the performance differences between all methods are small. However, our method still achieves about 0.4 % point promotion on 20-way 1-shot experiment, despiting that some methods fine-tune on the support set Vinyals et al. (2016); Finn et al. (2017); Li et al. (2017); Mishra et al. (2017).

## 4.3 *mini*IMAGENET

The *mini*ImageNet dataset is a subset of the large ILSVRC-12 dataset Russakovsky et al. (2015), consisting of 60000 image, with 600 images per class. In our experiments, the splits, introduced by Vinyals et al. (2016), are used where data is divided into three disjoint subsets: 64 classes for meta-training, 16 classes for meta-validation, and 20 classes for meta-testing. Following Santoro et al. (2016); Vinyals et al. (2016); Finn et al. (2017); Snell et al. (2017); Qi et al. (2018), we resize all images into $84 \times 84$ for training and testing.

**Influence of Number of Sub-generator**  The kernel generator is designed with a hierarchical structure to integrate information from multiple scales. In our experiments, there are 2 layers in the kernel generator, each layer for one sub-generator. Here, to investigate the influence of the number of sub-generator, we compare the results of model with 1 sub-generator and model with 2 sub-generators. Experiments are performed on the setting of 5-way 1-shot and 5-way 5-shot with the 4-layer network on *mini*ImageNet. The accuracies of model with 2 sub-generators increase from 53.15% to 54.47% and from 67.95% to 68.27% on 5-way 1-shot and 5-way 5-shot respectively relative to model with only 1 sub-generator, which indicates the effectiveness of hierarchical structure of the kernel generator.

**Influence of Distance Metric**   we evaluate three different distance metrics in the individualized feature space on *mini*ImageNet, including cosine similarity, Manhattan distance and Euclidean distance. The results are shown in Table. 2. As can be seen, the distance metric has limited influence on accuracies, and Euclidean distance performs slightly better on both setting of 5way-1shot and 5way-5shot.

**Visualizating of Individualized features**    As shown in Figure 3, we visualize the feature maps of the support set images in the individualized feature space corresponding to different query images. The first row shows three images $S_1$ to $S_3$ from support set. In each row of next 2-4, the leftmost image is the query image $Q_i$ used to generate kernels that map $S_1$ to $S_3$ into the feature space tailored for $Q_i$, and the rest images are the feature maps of $S_1$ to $S_3$ in $Q_i$'s feature space. Clearly, different aspects of a support set image are focused when mapped to distinct individualized feature spaces of different query images, demonstrating the rationality of our individualized feature space.

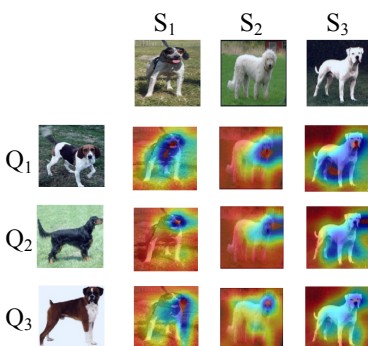

**Table 2.** Comparion of different distance metrics on *mini*ImageNet for 5-way 1-shot and 5-way 5-shot.

| Distance Metric | 5-way accuracy(%) | |
| --- | --- | --- |
| | 1-shot | 5-shot |
| cosine | 53.54 | 67.89 |
| Manhattan | 54.39 | 67.26 |
| Euclidean | 54.47 | 68.27 |

**Figure 3.** Illustration of feature maps in different individualized feature spaces.

**Table 3.** Few-shot classification accuracies on *mini*ImageNet. $C_4$ and ResNet are described in section 4.1.

| Model | Fine Tune | Networks | 5-way accuracy(%) | |
| --- | --- | --- | --- | --- |
| | | | 1-shot | 5-shot |
| Matching Nets Vinyals et al. (2016) | Y | $C_4$ | 43.56 | 55.31 |
| Meta-Learn LSTM Ravi & Larochelle (2017) | Y | $C_4$ | 43.44 | 60.60 |
| Prototypical Nets Snell et al. (2017) | N | $C_4$ | 49.42 | 68.20 |
| MAML Finn et al. (2017) | Y | $C_4$ | 48.7 | 63.11 |
| Meta-SGD Li et al. (2017) | Y | $C_4$ | 50.47 | 64.03 |
| Relation Net Sung et al. (2018) | N | $C_4$ | 50.44 | 65.32 |
| SNAIL Mishra et al. (2017) | Y | $C_4$ | 45.1 | 55.2 |
| SNAIL Mishra et al. (2017) | Y | ResNet | 55.71 | 68.88 |
| ours | N | $C_4$ | 54.47 | 68.27 |
| ours | N | ResNet | **56.89** | **70.51** |

**Comparison with State-of-the-art**    Here we compare our proposed method with other state-of-the-art approaches on the *mini*ImageNet data set. As is shown in 3, under both 5-way 1-shot and 5-way 5-shot setting, our accuracies are much higher than those metric learning based method, such as Matching Nets Vinyals et al. (2016), Prototypical Nets Snell et al. (2017), and Relation Net Sung et al. (2018), indicating the superiority of individualized feature space.

## 5    CONCLUSION

In this paper, we introduced a meta-learning method based on learning how to create feature embedding dynamically for different query images in the few-shot classification task. When recognizing a given query image, its characteristics are considered into contrusting the feature embedding, leading to an individualized feature space tailored for the query image, where the query image can be better distinguished from other objects. The meta-learner is designed as a kernel generator to create the dynamic convolution kernels. Good generalization of our models benefits from the kernel generator which can obtain the meta-knowledge of creating adequate convolutional kernels for different query images. The good performance on two data sets, i.e., Omniglot, and *mini*ImageNet, demonstrates the superiority of the proposed individualized feature space.

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
