# OpenReview forum: "Meta-Learning with Individualized Feature Space for Few-Shot Classification"
_ICLR.cc/2019/Conference_

### Official Review · AnonReviewer2 · 2018-11-01
**Interesting idea, but unclear contributions**

**Rating:** 3
**Confidence:** 3

**Review:**

This paper proposes a new meta-learner for few-shot learning that conditions the parameters of the model on the given query image. The authors argue that this allows the model to focus on features particular to the query, thereby facilitating classification. The paper introduces a kernel generator as a meta-learner and report performance on two standard benchmarks, Omniglot and miniImagenet.

Several methods propose meta-learners that adapt the learner’s parameters to the task or each class in the task. This paper adapts to the query itself, which may provide other benefits, and provides a useful complement to prior work on parameter adaptation in few-shot classification.

While the core idea itself is clearly articulated, the reading is dense and many of the finer points are vaguely presented. This makes the paper hard to read and its contribution unclear. In particular, the meta objective itself is not defined, the second loss function contains an undefined (learnable?) functions whose role is not entirely clear. In the experimental section, the authors mention that they use Prototypical Networks (Snell et al., 2017) on top of their kernel generator. This puts their contribution in a different light, now as an extension of Snell et al., (2017). I’m also unclear about the novelty of kernel generator the authors supposedly introduce. The kernel generator appears identical to that of Han et at. (2018), in which case the contribution is its application to few-shot learning, not the kernel generator itself.

Since the main contribution of this paper is to condition the learner’s parameters on the query, as opposed to the task or the classes in the task, the relevant comparison is with respect to such alternative methods. Several such benchmarks are missing (below), and when considered, the reported results are relatively weak.

For an up-to-date collection of benchmarks on miniImagenet, see Rusu et. al., (2018, https://arxiv.org/abs/1807.05960).

===

[1] Gidari and Komodakis. Dynamic few-shot visual learning without forgetting. 2018.
[2] Oreshkin et al.. TADAM: Task dependent adaptive metric for improved
few-shot learning. 2018.
[3] Qiao et al.. Few-shot image recognition by predicting
parameters from activations. 2017.
[4] Bauer et al.. Discriminative k-shot learning using probabilistic models. 2017.

---

### Official Review · AnonReviewer3 · 2018-11-03
**metric-based meta-learning via individual feature space embedding**

**Rating:** 5
**Confidence:** 4

**Review:**

Summary:
One of popular approach to few-shot classification is to learn an embedding function to a common feature space where the similarity between two examples is expected to be well determined. The current work claims that query-dependent feature space (referred to as individualized feature space) gains over the common feature space, in the task of few-shot classification. To this end, the paper employed a technique 'kernel generator' which has been recently proposed in [Han et al., 2018]. Few-shot classification is done using distance (e.g. Euclidean) in the query-dependent feature space.
The paper evaluates this method using Omniglot and miniImagenet.

Strengths:
- Constructing individualized feature space tailored to each query is a novel idea.
- The paper shows strong quantitative results.

Weaknesses:
- The clarity is a big obstacle in this paper. Section 3 contains the main idea on 'kernel generator' which is the critical technique to map input images to individualized feature spaces. Unfortunately Section 3 is hard to follow.
- Moreover, the idea of kernel generator is the one used in [Han et al., 2018], so the contribution of this paper is very limited.
- In a nutshell, the current work can be considered as a mix of matching network and kernel generator.

Specific comments:
- Regarding terminology, authors state that "there are three sets of examples in a few-shot classification task: a training set, a support set, and a testing set. The training set and the support set have disjoint label spaces with each other while the testing set shares the same label space with the support set." I am very confused with what authors mean by support set. In general, each episode has a support set as well as queries in both meta-training and meta-test phases. Meta-training and meta-test has disjoint label spaces.
- It is not clear to me what the problem setting is here. Queries in training and test phases have different label spaces. So, I am wondering feature space tailored to queries in the training phase can be well generalized to the test phase. Or you assumes that both cases have the same label space?
-Fig 2: The kernels and the conv features interact in a node which says “X”, making it seem like we are either pointwise multiplying or taking an outer product. The figure would be clearer if it somehow expressed that the two interact via convolution. (To add to this confusion, the kernels are thin which makes them look like vectors)
-eq(6): the index i is used to denote two things at once (g_i, c_ij). This notation should be different.
-eq(6): it says g_i is a fully-connected layer, but P_q^ij is a 3d tensor. Is g_i a 1x1 convolution, or do you flatten P?
-eq(9): what is H? Does it mean entropy? How is the set K_q a distribution of kernels? How does this loss relate to capturing the intrinsic characteristics of an object? This whole part should be clearer.

---

### Official Review · AnonReviewer1 · 2018-11-03
**Encouraging results but contribution seems incremental**

**Rating:** 5
**Confidence:** 4

**Review:**

# Summary
This work deals with few-shot learning and classification by means of similarity learning. The authors propose a method for generating a set of convolutional kernels, i.e. a mini-CNN, for a query image given a set of support samples (with samples from the same class and some other classes). Kernels are generated for each query and are adapted to the specific visual content found in the query image, thus a new embedding space is identified. The difficulty of the task is constrained by using a common base CNN for feature extraction, making the task resolution more feasible in the few shot regime. The method is evaluated on standard benchmarks Omniglot and miniImagenet with competitive results.


# Paper strengths
- The paper has a good coverage of related work

- The proposed method is interesting and the results are encouraging

- The authors argue and study the influence of multiple elements over their contribution: number of sub-generators, distance metric, choice of architecture

# Paper weaknesses
- My main concern with this work is the incremental contribution with respect to the work by Han et al. (2018), "Face recognition with contrastive convolution". In that work the authors proposed a convolutional kernel generator for every pair of images to be compared/matched, while here the principle is simplified to re-use the same convolutional kernels for the a query image. The loss functions are nearly identical, both works use a classification loss and a loss ensuring kernels at different images with the same object should be similar. The visualizations of the feature maps are similar as well, though these would have been necessary any way for this type of contribution.

- The architecture of the kernel generator is not clear to the reader. Is it similar with the one from Han et al.?

- The related and relevant work from Gidaris and Komodakis (2018), Dynamic few-shot visual learning without forgetting, is not included as baseline in the evaluation. Their method is superior when using C4 and similar (while still keeping performance levels on previous tasks).

- Given that the evaluation for few-shot classification takes random samples of query and support samples and that we're dealing with stochastic models, it's common and encouraged to include error-bars/standard deviations in the results to get a better idea on the performances. I encourage the authors to do the same.

- The visualizations from Figure 3 would need some additional clarifications from the authors in the text. It's not clear what does the colormap refer to, red is for high activation and blue for low activation (as typical for jet colormap) or the other way around? If blue is highly active, it's worrying that most dogs (Q1,Q2,Q3) are active on the white dog in S3. As said, it would be useful to have some comments from the authors in the text to better explain the visualizations

- Minor remarks:
    + The evaluation protocol from Omniglot should be specified as there are 2 ways of doing it: 1) using characters from different alphabets at test time (easier); 2) using characters from the same alphabet (more difficult)
    + There are some other works dealing with weight generation or with adaptive embedding that would be worth mentioning:
        * Y.X. Wang et al., Learning to model the tail, NIPS 2017
        * A. Veit and S. Belongie, Conditional similarity networks, CVPR 2017


# Conclusion
This paper advances an interesting idea for few-shot classification and gets competitive results. As mentioned in the section above, I'm worried about the incremental contribution on top of the work by Han et al.. In addition results are outperforming state of the art works, while requiring generating kernels and features for each query. My current rating is between Weak Reject and Borderline.

---

### Meta-Review · Area_Chair1 · 2018-12-14
**Good performance, but issues with clarity and novelty**

**Confidence:** 5
**Recommendation:** Reject

**Metareview:**

The reviewers all appreciate the idea, and the competitive performance, however the consensus is that this is a simple extension of the work of Han et al. and therefore the current submission contains little novelty. There are also numerous issues regarding clarity that the reviewers have pointed out. It is unfortunate that the authors have not engaged in discussion with the reviewers to resolve these, however they are encouraged to consider the reviewer feedback in order to improve the paper.